# Recent Progress of Fiber-Optic Sensors for the Structural Health Monitoring of Civil Infrastructure

**DOI:** 10.3390/s20164517

**Published:** 2020-08-12

**Authors:** Tiange Wu, Guowei Liu, Shenggui Fu, Fei Xing

**Affiliations:** School of Physics and Optoelectronic Engineering, Shandong University of Technology, Zibo 255049, China; wtg1103@stumail.sdut.edu.cn (T.W.); gwliu@stumail.sdut.edu.cn (G.L.); xingfei@sdut.edu.cn (F.X.)

**Keywords:** fiber-optic sensors, structural health monitoring, distributed fiber-optic sensor, optical time-domain reflectometer, civil engineering

## Abstract

In recent years, with the development of materials science and architectural art, ensuring the safety of modern buildings is the top priority while they are developing toward higher, lighter, and more unique trends. Structural health monitoring (SHM) is currently an extremely effective and vital safeguard measure. Because of the fiber-optic sensor’s (FOS) inherent distinctive advantages (such as small size, lightweight, immunity to electromagnetic interference (EMI) and corrosion, and embedding capability), a significant number of innovative sensing systems have been exploited in the civil engineering for SHM used in projects (including buildings, bridges, tunnels, etc.). The purpose of this review article is devoted to presenting a summary of the basic principles of various fiber-optic sensors, classification and principles of FOS, typical and functional fiber-optic sensors (FOSs), and the practical application status of the FOS technology in SHM of civil infrastructure.

## 1. Introduction

Modern large-scale civil engineering such as bridges, tunnels, space shuttles, large dams, and other infrastructure facilities have significant applications. These facilities not only require huge economic investment in the manufacturing process but also are closely related to personal safety. The facilities’ service life is usually decades or even hundreds of years [1,2]. In the course of using, they could inevitably be coupled with various disasters (such as environmental loads, fatigue effects, etc.). The structure could appear in different degrees and different kinds of damage [3,4,5,6], which may cause serious personal accidents and property losses. Therefore, whether these structures can work safely for a long time has attracted lots of attention. It is necessary to carry out real-time health monitoring and evaluation of engineering structures to put an end to the potential hazards and improve the safety performance of civil facilities.

Real-time SHM for major engineering structures can timely identify the cumulative damage of the structure and evaluate its service performance and life, and establish a corresponding safety early warning mechanism for early warning of possible disasters, which is not only of great scientific significance for improving the safety and reliability of the structure, but also can reduce the cost of operation and maintenance of the structure. It has become the inevitable requirement of the future engineering, and also a tough issue to be solved urgently [7,8,9,10,11,12,13,14].

SHM is an important application of intelligent material structure in practical engineering, which can monitor the “health” state of the structure on-line. It uses embedded or surface-bonded sensors as the nervous system to sense and predict internal defects and damage in the structure. The overall and local deformation, corrosion, brace failure, and other factors of the structure can be evaluated by the SHM system. When there is a sudden accident or dangerous environment, it can restore the whole structural system to the best working state through adjustment and control. Of course, the structure can protect itself and survive in times of danger by automatically changing and adjusting the shape, position, strength, stiffness, damping or vibration frequency of the structure.

In the process of SHM, the measured values of dynamic response sampled by the system are obtained through a series of sensors, the characteristic factors sensitive to damage are extracted from these measured values, and these characteristic factors are statistically analyzed to obtain the current health status of the structure. Generally speaking, civil engineering structures and major infrastructures have the characteristics of large volume, large span, wide distribution area, and long service life [15,16]. In order to realize the full-scale monitoring of strain, displacement, temperature, vibration, etc. of large-scale structures, usually dozens or even hundreds of sensors are required [17,18,19]. The SHM system could face the challenge of acquiring, transmitting, and storing large amounts of data. Recently, sensors have an important effect on various kinds of fields [20,21,22,23,24,25,26,27]. The commonly used sensors in the SHM system are piezoelectric element [28,29,30,31,32], strain element [33,34,35,36], and FOS [37,38,39,40,41,42,43,44,45,46,47,48,49,50,51,52,53,54]. Piezoelectric element can be used as both sensor and actuator with high sensitivity, good dynamic performance, and a wide range of applications, but it has some disadvantages, such as brittleness, not easy to be embedded in the structure, low-frequency characteristics, and so on. The strain element has the characteristics of high sensitivity, good static performance, and stable performance. However, the monitoring system composed of traditional resistive strain gauges is difficult to meet the needs of intelligent health monitoring of practical engineering structures in terms of performance stability, durability, and distribution range. In recent years, FOS has entered our field of vision with the development of material and structure damage signal processing technology and the deepening of sensing technology. With its advantages of small size, light weight, anti-corrosion, anti-EMI, and easy embedding, FOS has achieved rapid development and application all over the world [55,56,57,58,59,60,61].

## 2. Classification and Principles of Fiber-Optic Sensor (FOS)

The working process of the optical fiber sensor includes but not limited to the monitoring of external factors and signal transmission. When light propagates in an optical fiber, characteristic parameters such as light intensity, phase, polarization state, wavelength, or frequency will change. Therefore, FOS can be divided into intensity-modulated FOS, phase modulated FOS, polarization modulated FOS, and frequency modulated FOS. In the FOS system, the modulation occurs inside the optical fiber when the light in the optical fiber is transmitted from the light source to the detector, called the intrinsic FOS. The modulation that occurs outside the optical fiber called extrinsic FOS. FOS can be divided into interferential FOS and non-interferential FOS according to whether the light interferes or not. In addition, according to the induction range, FOS can be divided into point (local) FOS, quasi-distributed FOS, and distributed FOS. The most commonly used in civil infrastructure are Fabry–Perot fiber-optic sensor (FPFOS) [62,63,64,65], fiber Bragg grating (FBG) sensor [66,67,68,69,70,71,72,73,74], optical time domain reflectometer (OTDR) [75,76,77,78], and long-period fiber grating (LPFG) sensor [79,80,81].

### 2.1. Fabry–Perot Fiber-Optic Sensor

The optical fiber sensors utilize the change of optical phase affected by the physical field to reflect the properties of the measured objects. After being emitted by the light source, the light is divided into two beams with the same frequency, polarization direction, and initial phase through the prism. One beam is signal light and the other beam is reference light. Interference occurs when they meet. The interference image can be used to infer the influence of external factors on the signal light. Interferometric FOS can be divided into Michelson FOS, Mach-Zehnder FOS, Sagnac FOS, and Fabry–Perot FOS. The schematic diagram of the Fabry–Perot interference cavity is shown in Figure 1. We mainly introduce the FPFOS in this review.

The core of the FPFOS is the interference cavity. According to the different structure of interference cavity, FPFOS can be divided into intrinsic type and non-intrinsic type. In the intrinsic Fabry–Perot interferometric (IFPI) sensing system, the interference cavity is generally composed of a single-mode fiber-optic and an insulated mirror, and the end face of the fiber-optic cut by the fiber-optic can also be used as a mirror [82]. In the extrinsic Fabry–Perot interferometric (EFPI) sensing system, the interference cavity is composed of air or other non-fiber optic solid media [83,84,85,86]. The light source can use either a He-Ne laser or a low coherence light source to form interference after entering the optical Fabry–Perot cavity [83,84,85,86,87,88]. The above two types of Fabry–Perot FOSs make use of the physical parameters to be measured to cause the change of phase difference, so that the optical path difference is changed, and then the interference signal through the detector is transformed into electrical signal for processing. According to the change of coherent optical phase difference, the change of physical parameters to be measured can be obtained. The relationship between the phase difference of coherent light, the refractive index of the sensing fiber, and the cavity length is as follows:(1)$$\varphi =\frac{4\pi nL}{\lambda }=\frac{4\pi vnL}{c}$$
where *φ* is the phase difference of coherent light, *n* is the refractive index of the sensing fiber, *λ* is the wavelength of the light source, *L* is the cavity length of the resonant cavity, *ν* is the light frequency, *c* is the speed of light in vacuum [89].

The phase difference would change when affected by the strain of the sensing fiber, the frequency of the light source, and the temperature change of the environment. This effect can be quantified as:(2)$$\varphi =\varphi_{0}+\Delta \varphi_{L}+\Delta \varphi_{v}+\Delta \varphi_{T}$$
(3)$$\Delta \varphi_{L}=\frac{4\pi n}{\lambda }\Delta L=\frac{4\pi vn}{c}\Delta L$$
(4)$$\Delta \varphi_{v}=\frac{4\pi L}{c}(n+v\frac{dn}{dv})\Delta v$$
(5)$$\Delta \varphi_{T}=\frac{4\pi L}{c}(L\frac{dn}{dT}+n\frac{dL}{dT})\Delta T$$
where *φ* is the phase difference of coherent light, *φ_0_* is the initial phase difference, Δ*φ_L_* is the coherent optical phase difference lead by the strain of the sensing fiber, Δ*φ_v_* is the coherent optical phase difference caused by the light frequency, Δ*φ_T_* is the coherent optical phase difference result in the ambient temperature, *n* is the refractive index of the sensing fiber, *λ* is the wavelength of the light source, *L* is the cavity length of the resonant cavity, *ν* is the light frequency, *c* is the speed of light in vacuum, *T* is the ambient temperature. Assuming that the influence of ambient temperature on the phase difference of coherent light is negligible, and the light frequency of the light source remains unchanged, Equation (2) can be simplified as:(6)$$\varphi =\varphi_{0}+\Delta \varphi_{L}=\frac{4\pi n}{\lambda }L_{0}+\frac{4\pi n}{\lambda }\Delta L$$

In the Fabry–Perot fiber-optic sensing system, *R* is the reflectivity of the sensor mirror, the following relationship exists between incident light intensity *P_i_* and reflected light intensity *P_r_*:(7)$$P_{r}=2RP_{i}(1+\cos\varphi )$$

The coherent optical phase shift can be demodulated by detecting the change in the intensity of the reflected light, then the strain on the sensing fiber can be calculated, similarly obtaining the light frequency and ambient temperature.

### 2.2. Fiber Bragg Grating Sensor

SHM is the most active field in the application of FBG sensors. The low manufacturing cost, high-quality demodulation system, and practical packaging technology are important factors for the wide application of FBG sensors. FBG sensors can be attached to the surface of the structure or embedded in the structure to achieve real-time monitoring of the structure and monitor the formation of structural defects. Besides, a large number of FBG sensors can be connected in series to form a sensor network system, and the sensing signals can be remotely accessed to the central monitoring room for analysis and processing.

At present, the FBG sensor has become a commonly used sensor in the field of grating sensing [47,66,67,68,69,70,71,72,73,74,90,91,92,93,94,95,96]. The structure and principles of FBG are illustrated in Figure 2. When the broadband light source passes through the fiber grating, the narrowband spectrum is reflected. According to Prague’s law:(8)$$\lambda_{B}=2n\cdot \Lambda$$

In the Equation (8), $\lambda_{B}$ is the Bragg wavelength, *n* is the effective refractive index, and Λ is the grating period. When the measured physical parameters (such as temperature, stress, etc.) act on the fiber grating change, it could cause the change of *n* or Λ, which could lead to the drift of $\lambda_{B}$. On the contrary, the information of the measured physical parameters can be obtained by monitoring the drift of $\lambda_{B}$. The research work of Bragg grating is mainly focused on the quasi-distributed measurement of temperature and stress. The drift of $\lambda_{B}$ caused by the change of temperature and stress can be expressed as:(9)$$\Delta \lambda_{B}=2n\cdot \Lambda \{1-\frac{n^{2}}{2}[p_{12}-v(p_{11}+p_{12})]\}\cdot \varepsilon +2n\cdot \Lambda [\alpha +\frac{dn}{n\cdot dT}]\cdot \Delta T$$
where *ε* is the stress, *p_ij_* is the photopressure coefficient, *v* is the transverse deformation coefficient (Poisson’s ratio), *α* is the thermal expansion coefficient, and Δ*T* is the temperature change [13].

One of the most important FBG sensors is the quasi-distributed FOS based on FBG. Quasi-distributed FBG sensors use signal transmission fibers to connect multiple fibers or sensors together, and use the principle of multiplexing to separate the optical signals of different sensors, so as to analyze the monitoring data of different sensors. There is no need to lay the transmission fiber separately at each monitoring point compared with vibrating wire and resistive sensors, which makes engineering monitoring more convenient, low-cost, and efficient. Therefore, the multiplexed quasi-distributed FBG sensor is more suitable for monitoring crucial parts of large structures (such as pipelines, bridges, and dams), achieving multi-parameter measurement [97,98,99,100].

The wavelength signal of fiber grating sensor contains both temperature and strain information. To separate temperature information and strain information is a crucial issue in fiber grating sensor technology. Since 1993, people have been dedicated to studying the cross-sensitivity of fiber grating, many scholars have put forward numerous solutions. These solutions can be classified into: dual-wavelength transmission, dual-parameter method, temperature (strain) compensation method, fiber grating method with special performance, etc. In 2011, Stefani and Alessio [101] proposed a method for temperature compensation using adjacent gratings to the strain sensor, and gave a method for writing multiplexed gratings. In 2013, Ping Lu et al. [102] used the high-sensitivity outer layer mode to achieve temperature and axial strain compensation. In 2014, Yiping Wang et al. [103] improved the cavity length of FPI through repeated arc discharge to reshape the cavity, reducing the cross-sensitivity between tensile strain and temperature. In fact, in order to resolve the crosstalk between strain and temperature to distinguish the strain, temperature compensation is performed. Assuming that the two sensors would experience the same temperature change, a separate distributed temperature sensor is installed near the strain sensor. This makes the fiber strain-free and only sensitive to temperature.

### 2.3. Optical Time-Domain Reflectometer (OTDR)

The optical time-domain reflectometer (OTDR) uses the backscattered light of the fiber-optic to feedback the performance of the fiber-optic [75,78,104,105,106,107]. After the optical pulse emitted by the laser is injected into the fiber, the light energy received at the starting port can be divided into two types: (i) Fresnel reflected light of the fiber fracture surface or connection interface; (ii) Rayleigh scattered light. As shown in Figure 3, the detected backscattered light power returned at various points along the length of the fiber contains information about the loss suffered when the light is transmitted along with the fiber, so that the attenuation of the fiber can be analyzed and determined. With OTDR, we can measure the attenuation of the fiber, check the continuity of light, physical defects, or the location of the break, even measure the loss and position of the joint, and measure the length of the fiber. The Brillouin optical time-domain reflectometer (BOTDR) sensor is a classic classification of the FOSs and is based on the Brillouin scattering. Brillouin scattering is affected by temperature and strain, and the Brillouin spectrum produces frequency drift, which leads to stretching or compression in the axial direction of the fiber-optic. Therefore, the temperature and strain of the entire fiber can be obtained by calculating the frequency shift of the Brillouin backscattered light. OTDR unique single-ended monitoring technology has been widely applied in distributed monitoring of large-scale civil structures.

### 2.4. Long-Period Fiber Grating (LPFG) Sensor

Long-period fiber grating (LPFG) is a novel type of optical fiber passive device that has appeared in recent years, which forms a periodic or aperiodic distribution of refractive index in the fiber core. Because of the coupling effect of the internal field, the LPFG would reflect or transmit light of a specific wavelength. The long period of LPFG makes its resonance wavelength and amplitude extremely sensitive to ambient temperature, strain, bending, and torsion. Moreover, LPFG is a transmission type fiber grating with no backscatter and high measurement accuracy. Therefore, it plays an increasingly essential role in optical fiber sensing [108,109,110,111,112].

The LPFG couples the fundamental mode energy of the core to the cladding mode transmitted in the same direction. According to the coupling theory, the two coupled modes need to meet the following phase matching conditions:(10)$$\beta_{1}-\beta_{2}=\Delta \beta =\frac{2\pi }{\Lambda }$$
(11)$$\beta =\frac{2\pi n}{\lambda }$$
where *β* is the propagation constant of the mode, Δ*β* is the propagation constant difference of the coupled mode, Λ is the grating period, *n* is the effective refractive index, *λ* is the resonance wavelength. The above equations show the relationship between the resonance wavelength, the grating period, and the effective refractive index of the coupling mode. Thus, the resonance wavelength of the LPFG can be modulated. That is the change of the measured parameter can be detected by measuring the displacement of the LPFG resonance wavelength.

## 3. Typical Fiber-Optical Sensors (FOSs)

### 3.1. Crack Sensors

FOSs are extensively used in various fields [62,113,114,115,116,117,118,119,120]. The FOSs used for crack detection mainly including grating sensors and distributed fiber-optic sensors. Crack detection FOSs are mainly used for the stability of reinforced concrete structures. They have a wide range of applications in the health and stability of bridges, buildings, tunnels, and highways.

In crack detection, an important challenge is that it is difficult to monitor the number and depth of cracks in concrete structures due to the uneven and complex materials. Bao et al. [121] proposed a distributed crack fiber sensor based on optical time-domain reflection, which does not need to pre-determine the position of the crack, and realizes the coverage monitoring of all cracks. The short light pulse is used as the light source, and the backscattered light power is measured by OTDR. The formation of the crack is related to the bending angle of the optical fiber, and the bending leads to the loss of optical power. The relationship curve between backscattered power and light propagation distance decreases sharply at the crack. From this, the crack opening can be determined. One of the advantages of distributed optical fiber sensing is that it can monitor every point distributed along the optical fiber [122], thus it can accurately correspond to the location of the crack. Subsequently, on the basis of locating the crack position, Neha Niharika et al. [123] proposed a novel “S”-type optical fiber layout to increase the sensitivity while maintaining the distributed characteristics of the sensing system. Moreover, with the “S”-type fiber layout suggested by the solution, the sensitivity of crack openings is increased by 1.43 dBm/mm. In 2015, Gerardo Rodríguez et al. [124] demonstrated a method based on the optical backscattering reflectometer (OBR) to measure the generation, location, and width of cracks in concrete structures. A lot of uncertain structural damages are shown through cracks, thus the crack location and width are vital parameters. This OBR system can obtain continuous strain with higher spatial resolution and precision, and the experimental data calibrate the nonlinear model of the concrete slab, which can predict the crack location and width of different parts of the specimen. In 2018, Linked In and Yaming Li et al. [125] designed a new type of line crack sensor based on linear macroscopic bending loss of optical fiber. The sensor system overcomes the nonlinear relationship between macroscopic bending loss and crack opening displacement (COD), and verifies the simple linear relationship between macroscopic bending loss and COD of the optical fiber by using crack transfer device with gears.

### 3.2. Tempereture Sensors

In the past few decades, with the rapid construction of buildings, bridges, and dams, researchers have focused on the health monitoring of concrete structures. The temperature effect of the concrete structure is closely related to its structural health [126]. Temperature monitoring determines the quality, thermal resistance, and cold resistance of the concrete structure. At present, the more matured technology is the distributed optical fiber temperature sensing. Different from local optical fiber sensing, distributed sensing can realize the test of thousands of data points by a single sensor.

In the earlier period, Y. J. Rao et al. [127] proposed an optical fiber Fabry–Perot sensor based on wavelength multiplexing, which can be used to simultaneously measure the static strain, temperature, and vibration of SHM. It can be surface mounted or embedded to realize distributed temperature sensing. Different from most studies focusing on the influence of the surrounding environment on the temperature change of concrete structures, Xiaotian Zou et al. [65] designed a Fabry–Perot optical fiber temperature sensor to study the temperature change in the hydration process of concrete, and derived the temperature curve (as shown in Figure 4), which is used to calculate the apparent activation energy (Ea) and hydration heat (H(t)) of concrete, which can help us better understand the hydration of cement. When cement is mixed with water, an exothermic chemical reaction occurs to generate hydration heat [128]. The data obtained through experiments show that when the water-to-cement (w/c) ratio is 0.4, 0.5, and 0.6 respectively, the peak temperature of the concrete specimen is 51.42 °C, 52.88 °C, 55.08 °C. The early temperature changes caused by hydration heat at different water-cement ratios are related to the temperature stress and cracks of the concrete structure [129]. Therefore, during the hydration process, the temperature trend of the cement and the maximum temperature is crucial. These parameters can be used to predict future structural health.

In addition, distributed temperature sensing (DTS) plays an important role in controlling and monitoring the cracks in concrete structures. In large concrete structures (such as large bases, bridges, dams, etc.,), the hydration process heats the concrete structure after pouring large pieces of concrete. The outer surface of the concrete cools down faster than the inner surface. Because of the poor thermal conductivity of the concrete, a large temperature gradient is generated on the surface and inside. The uneven expansion of concrete caused by early temperature differences can cause thermal tensile stress on the surface. In the later period, it is constrained and deformed by the adjacent concrete or rock mass, which produces a tensile force on the constrained surface. When the tensile force exceeds the tensile strength, thermal cracking occurs [130]. Ouyang et al. [131] used the DTS system to provide crack control ideas for mass concrete structures in reservoir projects. This Raman-based DTS system usually consists of a DTS unit with an integrated OBR interrogation unit and multiplexer, computer, power supply, and optical cable. They are covered by a graded-index multimode optical fiber with a refractive index of 50 μm and a coating layer with a diameter of 125 μm, and the outer layer is covered with low-density polyethylene. Each fiber optic cable is connected to the DTS unit through a pigtail (Figure 5). The pigtail is a short length of optical fiber with a dedicated connector at one end to protect the fiber core wire and paired with the channel of the multiplexer; the other end uses a fiber fusion splicer to fuse it with the optical cable. Through the inverse analysis method based on temperature simulation, the temperature measurement value in the concrete block is used as the basic data to determine the thermal performance of the cast-in-place concrete. Based on thermal stress simulation using thermal characteristics, the cracking risk of each concrete block is predicted and evaluated in a temperature control mode related to time-varying construction and environmental condition, greatly improving the efficiency of temperature adjustment and crack control in the construction of mass concrete.

Although distributed temperature sensing technology has matured, the point sensors are more suitable for monitoring the temperature of a limited measurement point. The point FOS technology can be developed owing to the unique advantages provided by the use of optical fiber connecting the measuring location to the interrogating unit [132,133,134]. Moreover, low cost and mature packaging technology are also the advantages of point FOS widely used in temperature measurement in the industry.

### 3.3. Strain Sensors

In the SHM process, the density information in the structure can be used to identify the degree of deformation. In other words, when the strength of the structure is greater than the externally applied stress, the structure has higher stability. Therefore, it is important to identify the stress (load) or strength (damage) applied to the structure to ensure the health of the structure. In recent years, there have been many reports on the application of FBG and distributed fiber-optic sensor (DFOS) to structures performance monitoring, many of which are based on the FOS to measure the internal strain of structures.

In 2016, Sridevi. S et al. [135] reported an etched Bragg grating sensor (eFBG) coated with reduced graphene oxide (rGO), which significantly improved the sensitivity to strain and temperature because the interaction between the propagating light and the rGO film coated on the optical fiber is enhanced. In this study, the strain sensitivity of the eFBG sensor with rGO coating was 5.5 pm/με, which was about five times that of the bare FBG sensors and the resolution was 1με. The high aspect ratio, excellent flexibility, and ability to withstand strains up to 30% [136], as well as a higher temperature coefficient of resistance (TCR) than tungsten and platinum, make graphene suitable for manufacturing highly sensitive and durable strain and temperature sensor [137,138]. Because of its small size, FOSs can be placed on the surface of a structure or embedded inside a structure. When light propagates in an optical fiber, the transmitted and reflected light is modulated by its amplitude, phase, frequency, or polarization state. If the structure is affected by strain, then these parameters will change. The most commonly used FBG sensors and DFOS can’t provide multi-parameter sensing. Monssef Drissi Habtiet et al. [139] proposed to use a new type of sine wave sensor positioning to solve this problem. When the sensor is embedded inside the structure, the sinusoidal alignment model displays the multi-parameter strain more clearly than the linear model (Figure 6). When FOS is embedded in a large structure, it is difficult to identify multiaxial strain. If the direction of the applied strain is random, the linearly aligned FOS cannot distinguish the strain coordinates. Therefore, multi-axial strain with distributed FOS sinusoidal alignment in epoxy viscose fiber-reinforced composites is the best solution. Linearly arranged FOS can only detect transverse strain, while sinusoidally arranged FOS can provide linear, shear, and transverse strain information. As shown in Figure 7a, under similar boundary conditions, the strain value collected by the linearly aligned FOS is 1300 μm/m, and the strain value collected by the sinusoidally-aligned FOS is 600 μm/m. The strain range of linear alignment is 55% higher than that of sinusoidal structure. However, the advantage of sinusoidal alignment is that a shear strain of 100~250 μm/m (as shown in Figure 7b) can be detected when the torsional load is applied, so it can work under bending loads. Considering that the strain range should be close to the linear configuration of FOS while maintaining the torsional load strain sensing. The collected strain value reaches 1050 μm/m (as shown in Figure 7c) after extending the sinusoidal alignment period, which makes the difference between the strain values of the linear configuration and the sinusoidal configuration reduced to 20%. It is fully proved that it is possible to realize DFOS multi-parameter strain sensing without affecting strain.

Since 1998, Froggatt et al. [140] used optical frequency-domain refractometer (OFDR) to demonstrate distributed static strain measurement for the first time. The potential of its high spatial resolution in strain measurement has attracted widespread attention. OFDR uses a swept frequency laser interferometer to generate the relationship between strain or temperature and sensor length, with FBG or Rayleigh scattering as the source signal [141]. With the rapid growth of demand for dynamic disturbance measurement in the oil and gas, aerospace, and geophysics industries, OFDR’s method of realizing distributed vibration measurement has also been widely used. In 2015, Stephen T. Kreger’s team [142] developed and demonstrated a novel optical phase-based vibration detection and mapping technology based on the data of OFDR’s optical fiber sensing system. The result proves the potential of OFDR instrument for accurate, high spatial resolution, distributed vibration sensing in a dynamic environment, and is suitable for structural monitoring applications where modal frequency may be a health indicator. Since the optical fiber made of amorphous silica can be regarded as a naturally produced chaotic Bragg grating, the local reflection spectrum will also change with changes in strain or temperature [143]. Correlate the locally defined reference spectrum with the current spectrum to obtain the measured value of the frequency shift, from which the measured value (strain or temperature) can be derived. This kind of strain/temperature quantitative interrogation method has been proved in the SHM of civil, industrial, and aerospace structures.

Recently, femtosecond (FS) laser have attracted attention because of their extremely high peak power values, high spatial resolution, and ultra-short duration. Using FS laser to write gratings in optical fibers has quickly become a popular tendency [144,145]. Yinan Zhang et al. [146] proposed a FS laser micromachining method to manufacture a diaphram-based optical fiber Fabry–Perot interferometric (FPI) sensor for pressure measurement at high temperature. The sealed cavity of the diaphragm-based FPI sensor has an ultra-thin film (diaphragm) near the cutting optical fiber. The function of the diaphragm is to form interference as a mirror. The diaphragm would deform and alter the interference pattern when the environmental pressure changes. Therefore, the sensor can be used for pressure sensing.

The significance of utilizing FS laser is: (i) The laser polished surface helps eliminate the external reflection of the diaphragm surface, so that the sensor is not affected by changes in the refractive index of the environment; (ii) the cavity length of FPI can shorten to further reduce the cross-sensitivity to temperature; (iii) the thickness of the diaphragm can be controlled to meet the specific requirements for pressure sensitivity and measurement range. These advantages prove that the FS laser is an effective micromachining tool for manufacturing fiber optic equipment.

## 4. Applications of Fiber-Optic Sensor (FOS) Technology

### 4.1. Bridges

The life cycle of a large bridge is generally several decades or hundreds of years, and its life process generally includes planning and demonstration, design, construction, operation, management, maintenance, and repair stages. Because of the huge investment in the first two constructions, which are closely related to the health of the people’s travels and their importance, the overall planning of large-scale bridges is receiving increasing attention. At present, it is difficult to accurately predict and control because of the limitations of the understanding of complex bridge structures and the impact of natural disasters such as overdue service, corrosion, fatigue, impact, earthquake, and flood. In order to ensure the safety and durability of large bridges, it is necessary to understand their structural health in real-time. For the sake of comprehensive real-time monitoring of the performance of large-scale bridges during the operation phase, strengthening the maintenance of large-scale bridges, and then ensuring the safe and normal operation of the bridges and extending the life of the bridges, experts and scholars from various countries have carried out research on the real-time monitoring of bridge structures.

In 2017, Feng Xiao et al. [147] used the FBG inclinometer to monitor the dynamic response of the bridge. The inclinometer can not only record rotation or deflection but also monitor dynamic characteristics based on signal processing technology. Dynamic feature recognition is an important step in the monitoring of bridge operating conditions. If dynamic data (including natural frequency, vibration, and damping coefficients) are used to improve the damage identification of the bridge, it can provide more meaningful results [148,149,150]. Based on it, this study introduces a new idea to determine the frequency of large-span steel wall beams by monitoring the dynamic rotation angle of the expansion bearing. The FBG inclinometer can not only capture the natural frequency of the bridge but also provide rotation angle information. The FBG inclinometer monitoring system consists of sensors, multiplexers, interrogators, local computers, and remote computers, as shown in Figure 8. The FBG inclinometer is installed on the rocker bearing to monitor the rocking motion of the bottom roller. In the future, this inclinometer is expected to be installed on the deck or girder, while monitoring the rotation, and identifying the vertical dynamic movement of the bridge.

In 2018, Xiaowei Ye et al. [151] used FBG sensing technology to propose an orthotropic steel bridge stress monitoring program. In this work, the FBG sensors are deployed on the fatigue-prone rib-to-deck and rib to-diagram welded joints at the mid-span and quarter-span of the bridge, as shown in Figure 9. With the help of the wavelet multi-resolution analysis method, the local stress behavior under the influence of highway load and the temperature is analyzed. Further, through the method of finite mixed distribution and the parameter estimation method of genetic algorithm, the stress spectrum of rainwater count is modeled. The best probability distribution of the stress spectrum is determined by using the Bayesian information criterion (BIC). Besides, the thermal stress of the welded joint is calculated by the extrapolation method recommended by the International Welding Association. Figure 10 shows three finite mixed distributions of probability distribution function (PDF) and cumulative distribution function (CDF) for the stress range of the selected FBG sensor. The stress spectrum with the lowest BIC value is determined as the best probability distribution of the mixed normal distribution.

The state of the cable force of a suspension bridge is of vital importance to the safety of the bridge. For a suspension bridge that has already been built, the anchor cable structure of the bridge cannot be modified for cable force monitoring. Therefore, Dongtao Hu et al. [152] developed an FBG vibration sensor for online monitoring of the cable vibration characteristics of the Tongwamen Bridge. As shown in Figure 11, on the north and south sides of the bridge, FBG vibration sensors are installed symmetrically on 16 cables for distributed measurement. The cable vibration frequency is usually within 6 Hz. In order to avoid interference caused by the sensor’s resonance frequency, its frequency should be much greater than 6 Hz. The high frequency FBG vibration sensor made of traditional metalized packaging has high resonant frequency and low sensitivity, which is not suitable for cable testing. As shown in Figure 12, the FBG is fixed on the surface of the bridge, and the block object is fixed at the other end to obtain external acceleration and produce alternating bending strain on the surface of the beam. Then the strain is converted into wavelength information for demodulation. The experiment obtained a resonance frequency of 15 Hz, and the sensitivity was about 109.667 pm(m/s^2^), as shown in the Figure 13. Figure 14a,b show the dynamic force distribution of the cable monitored by 16 sensors on the south and north sides over time, respectively. It can be judged that the N7 and N8 cables on the north side are in a critical state and need to be repaired.

### 4.2. Buildings

In 2019, Aleksander Wosniok et al. [153] used DFOS to study the effect of static traffic loads on the slight deflection of existing bridge concrete structures. In this work, they tested the load on Amsterdambridge705 by using two 36-ton trucks parked at multiple locations on the bridge to record the longitudinal strain of the FOS embedded inside the bridge. The experimental results proved that the monitoring system supervised that the 93.9-m-long part of the optical fiber had detected a small elastic strain in the range of as low as 2 μm/m with a spatial resolution of 20 cm. Subsequently, Dong Yang et al. [154] proposed a deflection measurement for bridges based on the plastic optical fiber sensing (POFS) system. As shown in Figure 15, the system consists of three parts: a connecting tube for connecting to the measuring point, a liquid for filling the connecting tube, and a new plastic optical fiber liquid level sensor. The vertical deflection of the structure during the bending deformation of the bridge is judged by the change of the liquid level. The sensor shows a 1.9% change in the range of −5 to 40 °C, which is relatively stable. Its sensitivity is about 0.44 dB/mm of displacement, making the measurement result relatively accurate. However, further work is still needed to explore low-cost, high-sensitivity, and high-stability monitoring technologies.

Controlling the state of the load-bearing structure and building foundation of industrial facilities is an important part of ensuring safety. A. V. Tregubov et al. [155] have developed a novel type of DFOS for building temperature and strain measurement. The sensor uses an enhanced single-mode fiber as a composite optical element, which is placed in the glass fiber body. Compared with traditional FOS, ultra-high mechanical strength can better protect the sensor itself. The temperature sensitivity of the sensor is 0.1 MHz/deg, and the strain sensitivity is 2.4 MHz/mm. In the same year, K. Bremer et al. [156] developed two different optical fiber humidity sensors and optical fiber crack sensors in order to detect the effect of moisture on concrete structures. Among them, the distributed humidity sensor is an optical fiber sensor based on the expandable polymer of a polyvinyl alcohol hydrogel rod. Combined with OTDR technology, it is possible to determine the location of moisture in the space. The other is the FBG single-point optical fiber humidity sensor based on polyimide coating. After the coating material absorbs moisture and expands, the strain will be generated on the FBG sensor, and the strain level is linearly related to the relative humidity. Therefore, relative humidity information can be obtained by tracking the reflected Bragg wavelength. The crack fiber sensor is based on a single-mode fiber integrated textile mesh structure. This structure can transfer the elongation caused by the crack of the concrete structure to the optical fiber, so the optical fiber will break at the crack point, and the location of the crack can be determined by OTDR technology. In 2019, E. Badeeva et al. [157] developed an attenuating fiber-optic deformation sensor for monitoring the strain conditions affected by the geometrical disturbances inside the building and external bad weather, earthquakes, sudden temperature drops, and other factors. The use of FOS to monitor the state of buildings and structures can always keep improving until better, and there will be more work to conduct in-depth research and exploration in the future.

### 4.3. Tunnels

Recently, monitoring systems based on fiber-optic sensors have attracted great interest from researchers in the field of tunnel construction and optical engineering. Marcel Fajkuset al. [158] used BOTDR-based distributed optical fiber system (DSTS) to monitor the structural loads of highway tunnels. This work was carried out during the construction of a highway tunnel in Slovakia during a five-month long-term experimental measurement of tunnel load. Aldo Minardo et al. [159] reported a long-term DFOS-based on Brillouin scattering monitoring of railway tunnels affected by active earthflow, and the Brillouin frequency shift is used to monitor the strain (or temperature) distribution of the fiber. As shown in Figure 16, a laser diode with a wavelength of 1550 nm is divided into two beams as a light source. One beam serves as a probe beam, passes through the isolator, and is inserted into the probe to be connected to the fiber under test; the other beam is used to generate the pump, which successively passes through the electro-optic modulator (EOM), the erbium-doped fiber amplifier (EDFA), the polarization scrambler (PS), FBG, and the photodetector (PD). Brillouin scattering is an important physical parameter in fiber optics. Brillouin optical time-domain analysis is used to measure the strain or temperature distribution along the fiber. The probe optical signal and the frequency shift pulse pump are injected into the optical fiber separately. When the frequency of the two beams differs by a Brillouin frequency shift, interference occurs to generate strong sound waves. The sound wave generated is used as the diffraction grating and the pump fiber as the gain medium to amplify the probe light. The relationship between the gain and the frequency shift follows the Lorentz relationship, and the gain reaches the maximum when the frequency shift between the pump light and the probe beam is matched with that of the fiber grating. Therefore, the linear variation of Brillouin scattering with strain or temperature can be monitored by optical fiber. Piccolo A et al. [160] used DFOS technology, using Rayleigh scattering and finite element back analysis methods, taking advantage of different fiber anchoring methods on the circumference of the structure to monitor the convergence performance of the tunnel. The SHM of the shield tunnel is not mature enough and needs advanced sensing methods for monitoring. Tao Wang et al. [161] improved the existing DFOS and proposed the SHM solution for shield tunnels in operation. In order to solve the problem that the ordinary optical fiber is prone to breakage, they add a plastic tube to the middle part of the sensor to prevent the adhesion and deformation of FOS. The author conducted 55-day monitoring of the Nanjing Yangtze Shield Tunnel. Considering the long-term monitoring requirements, the durability of DFOS needs to be further improved.

### 4.4. Fiber Optic Temperature Sensing in Fire

In recent years, damage to buildings affected by the fire is not uncommon. The performance of the material is reduced due to fire, and the bearing capacity of the concrete structure is also greatly reduced [162,163]. Yi Bao et al. [164] used DFOS based on Brillouin scattering to measure the temperature distribution for the first time, and at the same time detected cracks in concrete structures during fire accidents. In the research, a pulsed pre-pump Brillouin optical time-domain analysis (ppp-BODTA) high-temperature sensor was used. Compared with traditional thermocouple or grating optical fiber sensors, the cost is lower and more data are collected in the same area. In the same year, Gorriz B et al. [165] designed a regenerative FBG innovative optical fiber sensor. The maximum temperature of concrete detected under fire conditions is 953 °C. The significance of this research is not only to confirm the feasibility of the optical fiber sensor working in the fire but also to provide further possibilities for exploring higher heat resistance concrete materials.

## 5. Improvements and Developments of Fiber-Optic Sensors (FOSs)

### 5.1. Development of New Fiber-Optic Technology

With the development of new fiber-optic technology, more and more newly invented FOS networks have optimized their performance. Professor Ole Bang and his team are dedicated to the research of microstructure polymer FBG sensing. Compared with traditional FOS, this sensor exhibits excellent performance in monitoring structural strain, temperature, humidity, etc., and has the advantages of high monitoring sensitivity and long-term stable operation [166,167,168]. Not only that, some advanced photonic technologies have also achieved excellent results in the field of optical fiber sensing. Demetrio Sartiano et al. [169] proposed an interrogation technique for cascaded FBG sensors based on microwave photonics techniques under coherent state. The novel technology is suitable for position and temperature measurement in the optical fiber of cascade write FBG. The impulse response is calculated by recording the electrical frequency response of the system, and the calculated impulse response is averaged to reduce noise and smooth the jagged appearance that is common in coherent measurements.

### 5.2. Forecast of Fiber-Optic Sensor (FOS) Development Direction

From a future perspective, several aspects may be considered to further develop the FOS-based SHM system. First, the performance of the FOS will continue to improve. For example, since the chemical and mechanical effects of concrete will reduce the mechanical properties of optical glass fibers, the long-term stability and reliability of FOS in a concrete environment will be explored in depth. In the bridge SHM system, a real-time, effective, robust, easy-to-install and collected data search and monitoring system will be further developed, which is a highly sensitive and repeatable DFOS suitable for cable frequency detection. In addition, the temperature, strain monitoring, and anti-noise performance of the FOS will be further improved in the tunnel SHM system. Second, in order to improve the stability and sensitivity of this type of monitoring technology and cost reduction, further work is needed, such as exploring advanced sensor production technology and the application of new POFS materials. Third, combining a variety of technical methods to further study the FOS and expand their scope of application. For example, the combination of DFOS and inverse analysis-finite element methods also represents an innovation in the field of SHM. Fourth, the cross-sensitivity of strain and temperature exists in most FOSs. FBG sensors are widely used in SHM, and the strain and temperature will cause the reflected Bragg wavelength shift. Therefore, the two factors must be separated to accurately measure each variable. Fifth, during the use of the FOS, in terms of maintaining accuracy and accuracy, the calibration of the sensor must be considered, and the changed sensor index must be corrected in time. We believe that in the future, more and more researchers will continue to work hard to solve every problem and achieve new development in the field of structural health testing.

## 6. Conclusions

In the past few decades, since SHM has come into our sight, it has been an important direction in the development of large-scale civil engineering. The emergence of new technology brings not only function and convenience, but also technical improvement and problems. In recent years, the development and application of optical fiber sensing technology in the field of SHM are more and more mature and stable. In this review, the working principles of FPFOS, FBG sensor, OTDR and LPFG sensor are introduced, and the distributed fiber-optic sensing technology is widely discussed and reviewed, especially in civil engineering structure. Then several classical functional sensors in civil engineering are described, including crack sensors, temperature sensors, and strain sensors. After that, the latest applications of different FOS in large-scale civil engineering such as bridges, buildings, and tunnels are reviewed. These works are related to the design of the sensor, the implementation technology, experimental results, and sensor performance. In addition, we keep eyes on the development of new fiber-optic technology. Also, we briefly summarize the difficulties faced by FOS in the field of SHM and predict its future development direction. There is still a lot of work to be done if FOS is to become a comprehensive, definite, and high-level feasible solution in SHM applications.

## Figures and Tables

**Figure 1 sensors-20-04517-f001:**
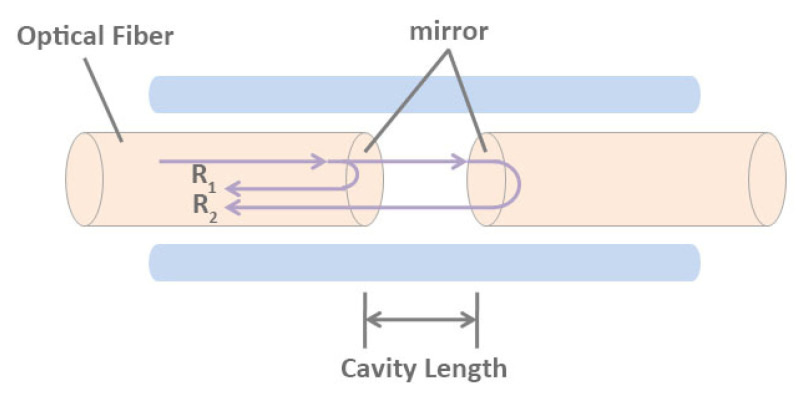
Schematic diagram of Fabry–Perot interference cavity.

**Figure 2 sensors-20-04517-f002:**
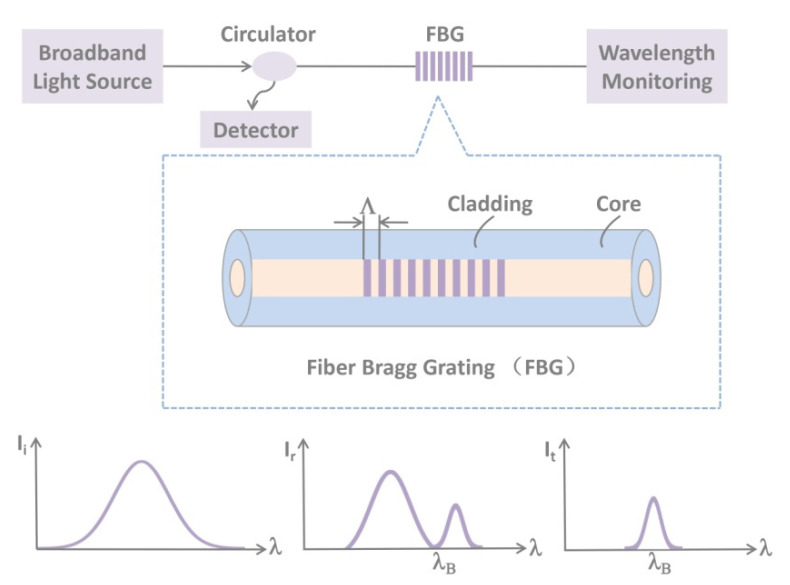
The principles and wavelength shift of fiber Bragg grating (FBG) sensors.

**Figure 3 sensors-20-04517-f003:**
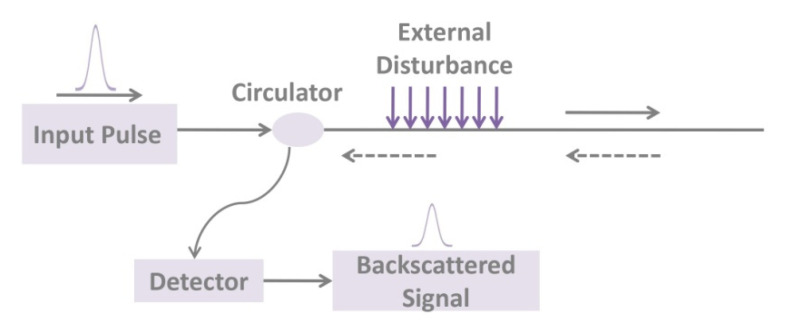
The principle of optical time-domain reflectometer (OTDR) based on backscattering.

**Figure 4 sensors-20-04517-f004:**
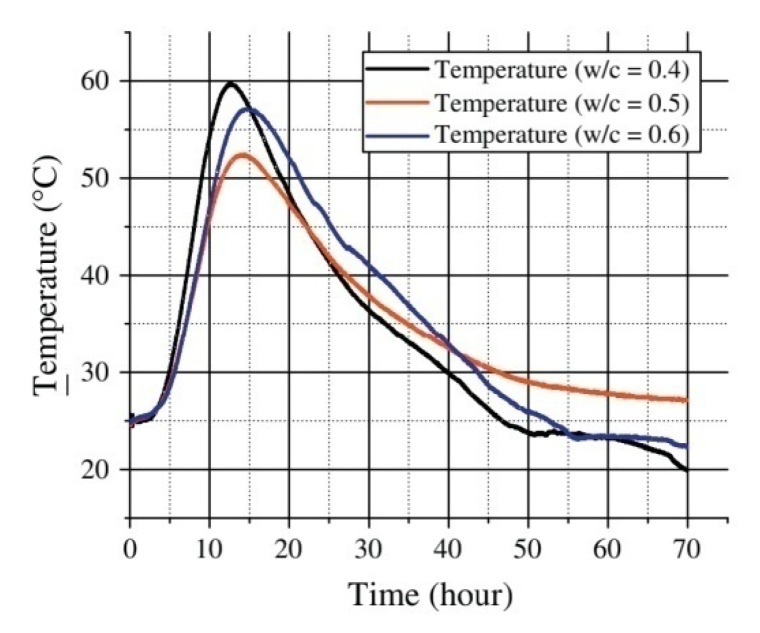
Concrete hydration experiment with water versus cement ratio 0.4, 0.5, and 0.6 using the thermocouple [65].

**Figure 5 sensors-20-04517-f005:**
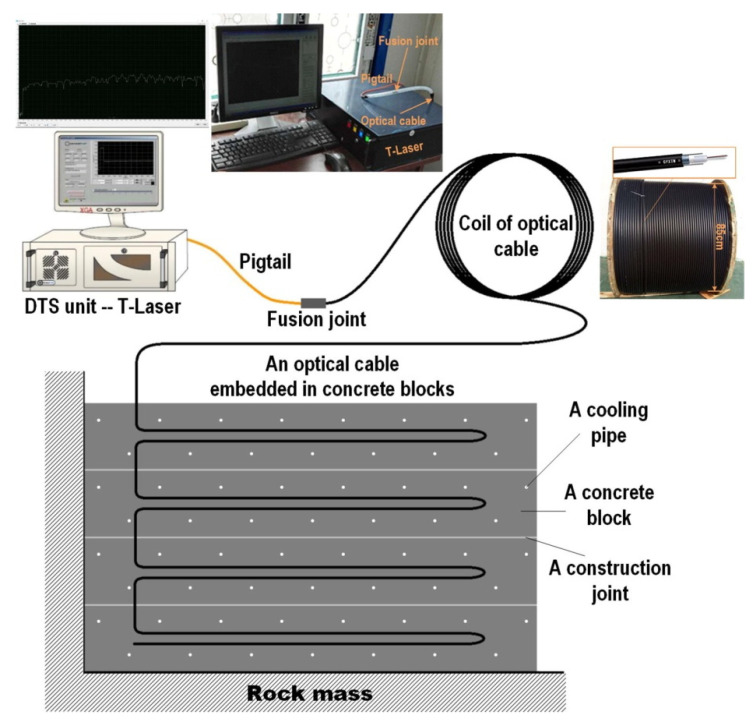
The diagram of the distributed temperature sensing (DTS) system [131].

**Figure 6 sensors-20-04517-f006:**
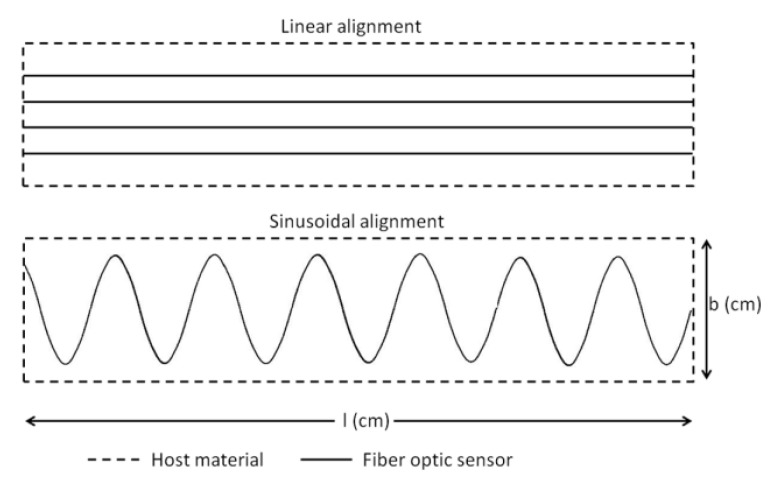
Fiber-optic sensor (FOS) installation method for a reference surface area [139].

**Figure 7 sensors-20-04517-f007:**
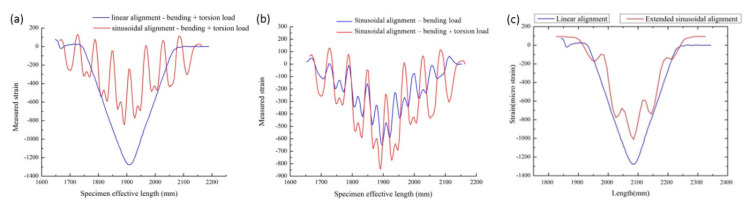
(**a**) Linear vs. sinusoidal alignment, bending + torsional load; (**b**) sinusoidal alignment, bending load with torsion vs. without torsion; (**c**) linear vs. extended sinusoidal alignment, bending load [139].

**Figure 8 sensors-20-04517-f008:**
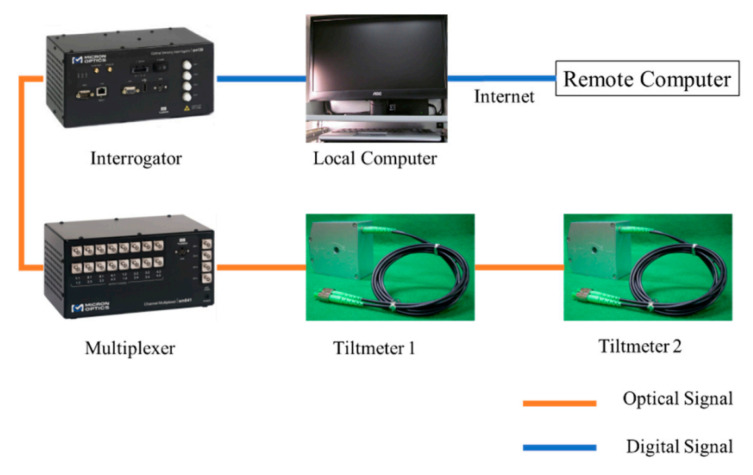
Configuration of the fiber Bragg grating (FBG) inclinometer monitoring system [147].

**Figure 9 sensors-20-04517-f009:**
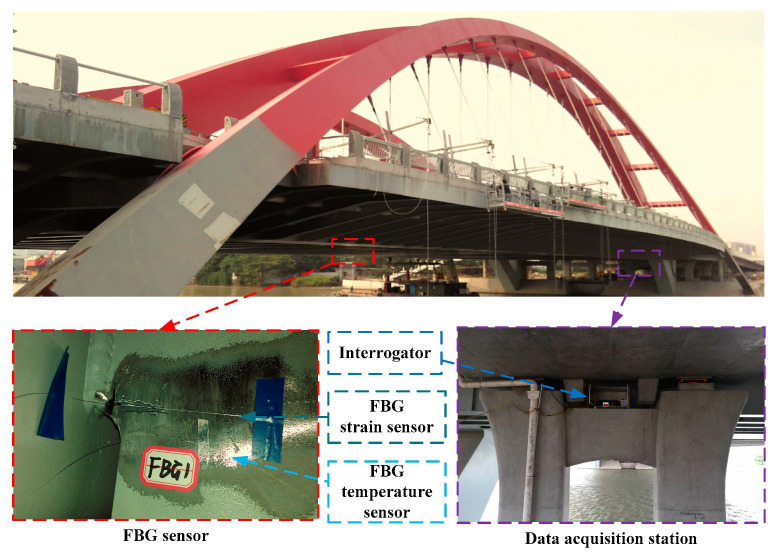
Deployment of structural health monitoring (SHM) system based on fiber Bragg grating (FBG) [151].

**Figure 10 sensors-20-04517-f010:**
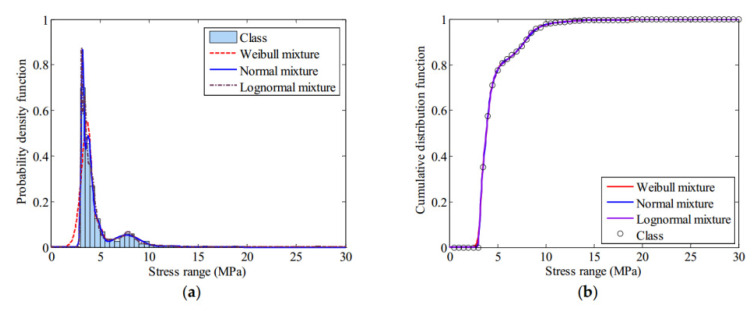
Distribution function of stress range: (**a**) finite mixture probability distribution function (PDF); (**b**) finite mixture umulative distribution function (CDF) [151].

**Figure 11 sensors-20-04517-f011:**
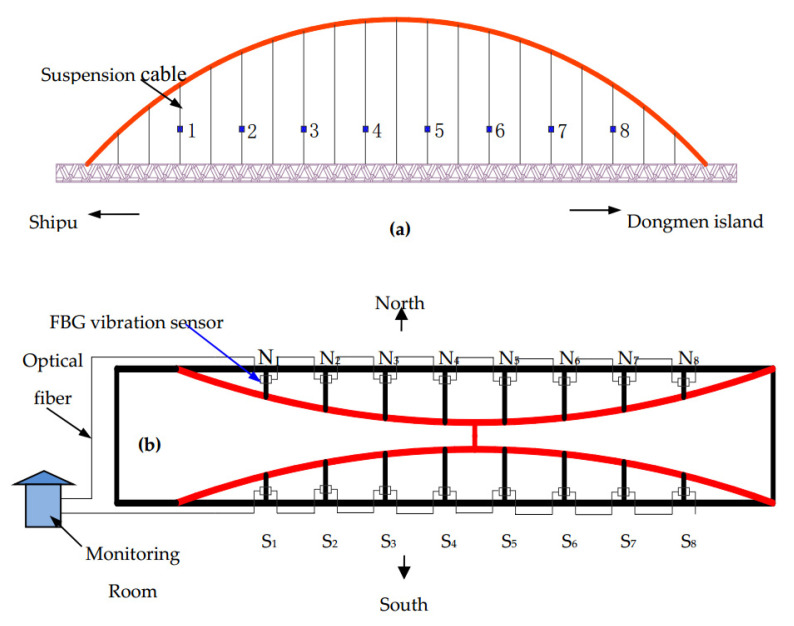
Arrangement of fiber Bragg grating (FBG) vibration sensors on Tongwamen bridge cables: (**a**) side view; (**b**) overhead view [152].

**Figure 12 sensors-20-04517-f012:**
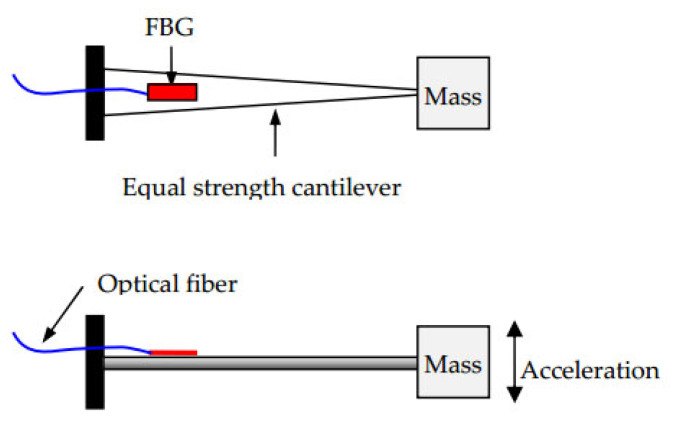
Structural schematic diagram of the designed fiber Bragg grating (FBG) sensor [152].

**Figure 13 sensors-20-04517-f013:**
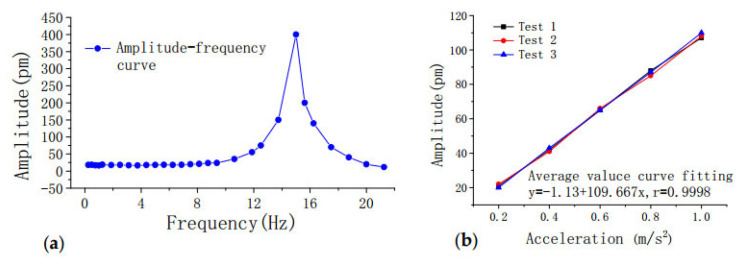
Performance test of the fiber Bragg grating (FBG) vibration sensor: (**a**) amplitude-frequency curve; (**b**) acceleration characteristics curve [152].

**Figure 14 sensors-20-04517-f014:**
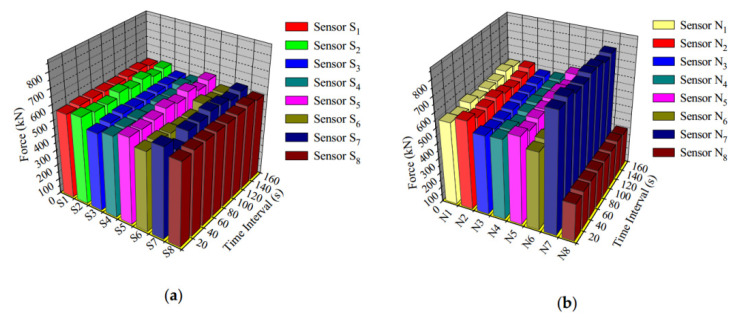
Suspension cable force distribution data measured by fiber Bragg grating (FBG) sensors: (**a**) south side suspension cables; (**b**) north side suspension cables [152].

**Figure 15 sensors-20-04517-f015:**
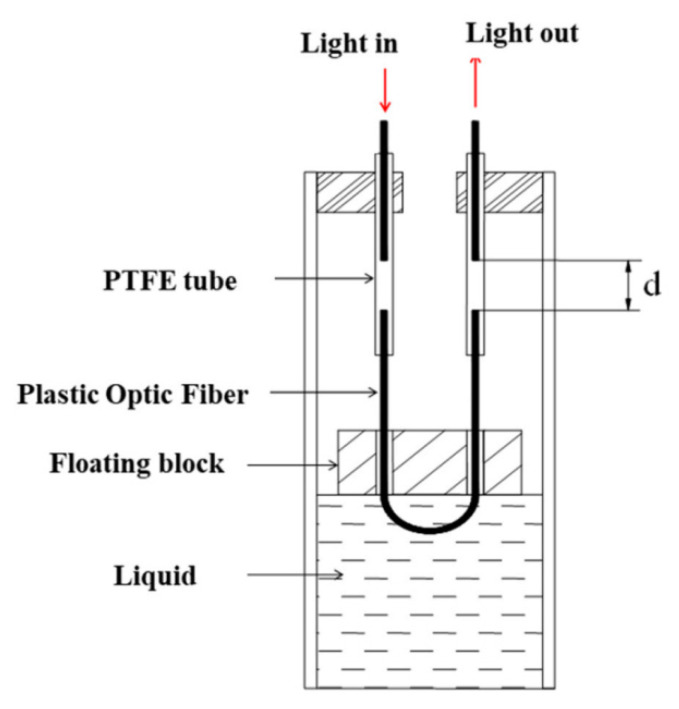
Fiber-optic-based liquid sensor setup. Note: PTFE = polytetrafluoroethylene [154].

**Figure 16 sensors-20-04517-f016:**
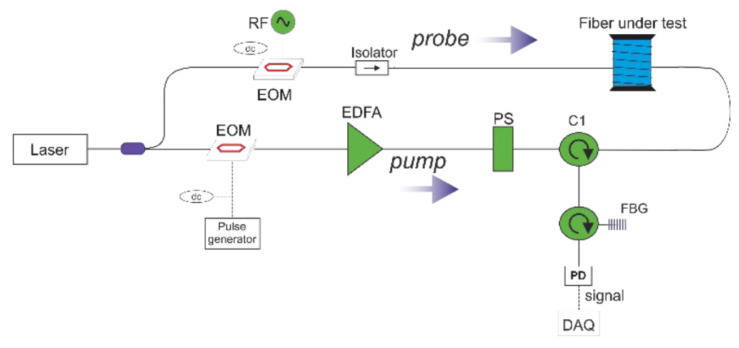
Experimental setup for distributed measurement of temperature in optical fibers. EOM: electro-opticmodulator; PS: polarization scrambler; EDFA: erbium-doped fiber amplifier; PD: photodetector; FBG: fiber Bragg grating [159].

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
