# Peer review of "Recent Progress of Fiber-Optic Sensors for the Structural Health Monitoring of Civil Infrastructure"

_sensors, 2020, doi:10.3390/s20164517_

Round 1
Reviewer 1 Report
Each manuscript section begins with its own introduction. It seems, that the whole manuscript consists of introductions of different articles.
Pictures from the manuscript are primitive, they do not explain features of described methods. The manuscript looks like a first chapter of a bachelor or a master dissertation.
My comments are below:
- The percentage of old articles (47.5%) in the references list is very large. That is why the manuscript cannot be named “recent progress”.
- The references No: 1, 2, 3, 4, 5, 6, 11, 12, 13, 14, 17, 26, 27, 35, 37, 38, 39, 40, 41, 43, 45, 46, 47, 50, 51, 53, 54, 56, 58, 60, 61, 64, 65, 66, 73, 77, 78, 79, 80, 81, 82, 83, 84, 85, 86, 87,88, 90, 91, 93, 94, 95, 99, 100, 101, 102, 103, 104, 105, 106, 107, 108, 109, 110, 114, 115, 140, 141 are published more than 10 years ago.
- The references No: 7, 8, 36, 42, 44, 48, 49, 52, 75, 118, 135 are not dated. It is not admissible.
- Line 101. This equation is very well known, but there is no external reference.
- Lines 110-113. It is necessary to give a definition for all variables which are used in the equations (2)-(5).
- Lines 114-116 after equations (2)-(5). A reasonable way or method to differ each term contribution into the equation (2) is not given.
- Lines 118-125. The paragraph contains good statements, which are very common, and can be applied for any type of sensors.
- Lines 127-128. It is a doubtful sentence. It is the refractive spectra under the influence of ambient temperature that changes, not refractive index.
- Line 133. "and" must be changed on "or" in "...the change of n or Λ,...", because the Λ changes mainly at constant n.
- Line 138. There is no external reference on the equation (7).
- There are mistakes in Figure 2. 1) It is necessary to replace "Coupler" by "Circulator". 2) The fiber Bragg grating spectra cannot have a shape of a triangle. It must be a Gaussian form at least. 3) There are no axis captions 4) There are generic scheme symbols for denoting laser source, photodetector, circulator, etc.
- Nothing is said about sensor multiplexing in a quasi-distributed sensors system.
- There are mistakes in Figure 3. 1) It is necessary to replace "Coupler" by "Circulator". 2) the signal shape cannot be a rectangle, it must have a Gaussian shape at least. 3) There are generic scheme symbols for denoting laser source, photodetector, circulator, etc. 4) It would be better to use a presentation of a frequency shift of the Brillouin backscattered light scheme instead of Figure 3. 5) The backscattered signal form is wrong. Only backscattered signal form (in time and in frequency domain) carries information. It cannot be the same as the input pulse.
- Lines 163-222. This piece of the manuscript looks as an introduction. Each section of the manuscript has its own introduction. It looks like the manuscript was combined from independent articles.
- Lines 163-222. Nothing is said about a required measurement range. There are common words only.
- Lines 223-276. Section 3.2. Table 1 has nothing common with the manuscript title and its abstract.
- Line 248. The mentioned temperature curve will be more interesting instead of the table 1.
- Lines 252-274. Nothing is said about point sensors. This paragraph does not contain any useful or interesting information.
- There are mistakes in Figure 4. It is not "a schematic of the DTS", this picture does not explain a DTS principle. The picture must be replaced.
- Lines 278-301. Section 3.3. This part of the manuscript looks like an introduction and does not carry any significance.
- FOS installation method is good, but it is well known, and it is not exclusive. It is not clear why the authors cited only this method.
- Figures 5 and 6 duplicate each other. Only one of them is enough.
- There is a mistake in the Figure 7. What can this figure explain? There are several photos with lines between them without any meaning.
- Lines 308-377 are the description of bridge health monitoring. This part of the text looks like an introduction, also.
- All Figures in section 4.1 are taken from https://doi.org/10.3390/s18020491. Many of them (Figures 8, 9, 10, 11, 12 and 13) are in weak correlation with main manuscript theme, namely (from abstract): "The purpose of this review article is devoted to presenting a summary of the basic principles of various fiber-optic sensors, classification and principles of FOS, typical and functional fiber-optic sensors (FOSs), and the practical application status of the FOS technology in SHM of civil infrastructure".
- There is no description of the scheme presented in Figure 15.
- Lines 445-457. Section 4.4 does not contain any useful information except references on external works, which anybody can find himself. Two references are older than 10 years.
- The conclusions of the manuscript in part "For more detailed information, it is recommended to read the corresponding references. In addition, we briefly summarize the difficulties faced by FOS in the field of SHM and predict its future development direction." gives an excellent assessment of the work. Besides, I did not find any predictions.
- Nothing is said about sensor multiplexing in a quasi-distributed sensors system.
- Nothing is said about a calibration of sensors.
- Nothing is said about a temperature compensation methods.
Reviewer 2 Report
This review paper presents a summary of the basic principles of various fiber-optic sensors (FOSs), classification and principles of FOSs, typical and functional FOSs, and the practical application status of the FOS technology in structure health monitoring (SHM) of civil infrastructure. Overall, the manuscript covers a broad range of FOSs (i.e., Fabry-Perot interferometer (FPI), fiber Bragg grating (FBG), optical time-domain refractometer(OTDR)) and applications (i.e., bridges, buildings, and tunnels), which is very useful for the researchers working in this area. However, I need to reconsider it for the publication after addressing the following questions.
1. I would suggest the authors do a more comprehensive literature review in this area. Here are several examples listed below:
a) Intrinsic Fabry-Perot interferometric sensor
Weak reflectors can be fabricated inside the fiber using a powerful tool(i.e., ultrafast laser) and then the device can be used as a distributed sensor for temperature or strain sensing.
b) Long period fiber grating
Long period fiber grating sensor has been proposed and researched for SHM in recent years, but I did not see any references here.
c) Special fibers
Polymer fibers has been researched for a while, please check publications in Ole Bang`s team.
d) Coaxial cable
Compared with FOSs, coaxial cable would be another candidate for SHM, what`s pros and cons for this new candidate?
3) OFDR and microwave photonics
Besides OTDR, a lot of people are working on optical frequency-domain refractometer (OFDR) or microwave photonics for sensing applications.
2. For the SHM, there might be a cross-talk issue between temperature and strain sensing, how to deal with this issue in the literatures?
3. Typos and misleading
Several typos have been found, i.e., Eq.4, .... the working principles of FPFOS, FBG sensors, and OTDR sensors are introduced...
Based on my understanding, OTDR means optical time-domain refractometer.
4. How to define recent process in this manuscript? 5 years? 10 years? I would suggest the authors focus more publications in the recent 5-10 years, not the ones published back to 1997 or 2003.
Round 2
Reviewer 1 Report
The manuscript became better than it was.
Reviewer 2 Report
The revised version is much better than V1 and could be considered to publish in Sensors.